# Ukrainian Women's Entrepreneurship and Business Ecosystem during the War: New Challenges for Development

**Tetiana Stroiko** [1] , **Patricia P. Iglesias-Sanchez** [2,*] , **Carmen Jambrino-Maldonado** [2] , **Elena Fernández-Díaz** [2] **and Carlos de las Heras-Pedrosa** [3]

1 Department of Economics, Management and Finance, V.O. Sukhomlynskyi National University of Mykolaiv, 54030 Mykolaiv, Ukraine; tanyastroyko@gmail.com

2 Department of Business Administration, Universidad de Málaga, 29071 Malaga, Spain; mcjambrino@uma.es (C.J.-M.); efernandezdiaz@uma.es (E.F.-D.)

3 Department of Audiovisual Communication and Advertising, Universidad de Málaga, 29071 Malaga, Spain; cheras@uma.es

\* Correspondence: patricia.iglesias@uma.es

**Abstract:** This research focuses on the key aspects of the basis of women's entrepreneurship in the particular case of the Ukrainian ecosystem of entrepreneurship. Even in wartime, entrepreneurship is a chance to overcome circumstances, and it should be developed for women and men. A correlation regression analysis and simulation modeling were carried out in order to rank the priorities for the development of types of economic activity of small business entities managed by women, making it possible to substantiate the forecast of the development of women's entrepreneurship in Ukraine for 2024–2026. The findings show the positive impact of the entrepreneurial ecosystem and the need to continue to strengthen policies that encourage and facilitate female entrepreneurship. This analysis provides three types of development scenarios for women's entrepreneurship in Ukraine.

**Keywords:** entrepreneurship; women's entrepreneurship; business ecosystem; institutional environment; barriers/non-barriers for women's entrepreneurship

## 1. Introduction

Overcoming significant gender disparities in labor force indicators, namely, increasing the participation of women in labor activities, can be an effective way of increasing the rate of economic growth and the standard of living of the population [1–7]. According to previous research, this positive effect occurs in countries with different levels of economic development.

Women's entrepreneurship is growing all over the world, and this phenomenon also has an impact on economic growth and well-being [8,9]. In fact, the development of women's entrepreneurship is increasingly seen as a significant factor in economic growth, job creation, income equality, and social inclusion [10,11].

Despite positive progress, women are disadvantaged and underrepresented as entrepreneurs [12]. According to the Global Entrepreneurship Monitor [13], women have been disproportionately affected by the pandemic, taking on a greater role in caring for relatives and in home schooling; thus, the gender gap in entrepreneurship is likely to have widened. For example, only 31 percent of self-employed professionals are women in the EU [14]. Men started a new business more often than women in 2022 [15]. According to the latest report by the Global Entrepreneurship Monitor [15], of the 49 countries participating in the corresponding study, there were only 4 in which the rate of new female entrepreneurship exceeded the rate of male entrepreneurship.

Recognition of women's entrepreneurship as a source of new jobs for both female business founders and other workers is growing in many countries around the world. In addition, women entrepreneurs can provide society with a variety of views and approaches

to management, enterprise organization, and solving business issues [16,17]. The benefits for the economy and society provided by an active policy on the development of women's entrepreneurship are multifaceted and significant [18,19].

Women's entrepreneurship in Ukraine has gone through a difficult developmental period. The period of formation of entrepreneurship has been studied in detail from a historical point of view [20]. The experience of female entrepreneurship and the peculiarities of its formation in the countries of Central and Eastern Europe have been studied, with the exception of Ukraine, despite its similarities with the countries of Eastern Europe [21].

The study of barriers faced by Ukrainian women was systematized in a joint US–Ukrainian study based on the phenomenological method of entrepreneurs [22]. The development of entrepreneurship in the post-Soviet countries has become a new economic force, and women play an important role in this [23,24]. However, the impact of war on entrepreneurship, and on women's entrepreneurship in particular, is currently underrepresented in the literature.

Therefore, it is important to study the stability of women's entrepreneurship during the war as well as forecast the development of women's entrepreneurship as an important component of post-war reconstruction of Ukraine.

This research paper is divided into three main sections. The first section reviews the existing literature and contextual data on women's entrepreneurship in Ukraine. The next section explains the methodology, followed by the sample and instruments used, etc. The last section contrasts the hypothesis and the proposed three forecast scenarios, combining descriptive analysis and a multivariate statistical analysis. This will allow us to draw conclusions and look into the practical implications for decision making regarding the challenges of recovery for Ukraine.

This paper ends by identifying the main limitations and proposing future lines of research. The main contribution is the discovery that the Ukrainian business ecosystem is a factor in women's entrepreneurial development and a driving force for maintaining stability and integrity in the conditions of an aggressive external environment. In the long run, this will create prerequisites for improving the economic and social well-being of society after the end of the war.

## 2. Literature Review

### 2.1. The Entrepreneurial Ecosystem as a Factor in Development

Recently, many studies have been conducted on the influence of the external environment on the functioning and development of business structures [25], taking into account the influence of the external environment on entrepreneurship, focusing attention on the following four factors: entrepreneurial orientation, environmental conditions, resources (human, financial, and social), and the owner's attitude toward development processes.

According to Robinson and Acemoglu [26], De Long and Summers [27], Ngepah [28], and Gupta, Pouw, and Ros-Tonen [29], different countries are looking for recipes to create effective economic institutions, capable of activating the business environment, to increase welfare.

An effective entrepreneurial ecosystem can become one of the basic driving forces of economic recovery and development in the future [30], and, when transforming under the influence of aggressive external factors, this ecosystem becomes more stable and adaptive [31].

The case of Ukraine is especially interesting because Ukraine's economy has proven that it is capable of sustaining a blow and adapting to significant force majeure circumstances. Since 2014, Ukraine's economy has functioned under the conditions of Russia's hybrid war against Ukraine. At the same time, with the national economy having shown relative stability, it quickly moved from a phase of crisis to a phase of restorative growth. In particular, during 2016–2019, the GDP of Ukraine increased by 12%. Throughout 2021 and early 2022, the economy was overcoming the consequences of the COVID-19 pandemic.

According to the Inclusive Development Index (IDI) of 2018, published by the World Economic Forum, Ukraine ranks 49th among 74 countries evaluated. According to this index, which assesses the degree of prevalence of the positive effects of the economic growth of the global population, Ukraine belongs to the category "emerging economies".

In the modern world, the quality of the institutional environment can be assessed using international rating systems, such as Doing Business, the Index of Economic Freedom, The Global Innovation Index, and the European Innovation Scoreboard, which currently have the largest coverage of the countries of the world. Each of these systems models the institutional environment of the national economy, highlighting the most significant factors (Table 1).

**Table 1.** Ukraine in global rankings (2020–2022).

| Property rights 39.7 | Judicial effectiveness 31.4 | Government integrity 33.8 | Tax burden 89.1 | Government spending 44.5 | Fiscal health 73.6 |
|---|---|---|---|---|---|
| Index of Economic Freedom (2022—184 countries) Ukraine—130th in world rankings; 44th in regional rankings; 54.1—overall score | | | | | |
| Business freedom 61.6 | Labor freedom 60.7 | Monetary freedom 71.2 | Trade freedom 78.6 | Investment freedom 35.0 | Financial freedom 30.0 |
| Starting a business 61 | Dealing with construction 20 | Acquiring electricity 128 | Registering property 61 | Getting credit 37 | |
| Doing Business (2020—190 countries) Ukraine—64th in world rankings; 70.2—overall score | | | | | |
| Protecting minority investors 45 | Paying taxes 65 | Trading across borders 74 | Enforcing contracts 63 | Resolving insolvency 146 | |
| Institutions 97 (places) | Human capital and research 49 | | Infrastructure 82 | Market sophistication 102 | |
| The Global Innovation Index (2022—132 countries) Ukraine—57th in world rankings; 34th in regional rankings; 57.0—overall score | | | | | |
| Business sophistication 48 | Knowledge and technology outputs 36 | | Creative outputs 63 | | |

Sources: Tyrrel and Kim [32]; The World Bank [33]; WIPO [34].

According to the Index of Economic Freedom, 2022 [32], produced by the American Heritage Foundation, Ukraine fell three positions, taking 130th place among 177 countries (in 2021, Ukraine was in 127th place). In the European region, Ukraine was placed between Russia and Belarus, occupying the penultimate 44th place. Although Ukraine has recorded an impressive overall increase in economic freedom amounting to 6.0 points since 2017, thanks to increased labor and financial freedom, it is still in the middle ranks of "mostly unfree" countries.

According to the results of Doing Business (2020) [33], Ukraine took 64th place. This was the best result for Ukraine during the entire study period. Since these ratings began to be calculated in 2006, Ukraine has not occupied the top positions. Only in 2015 did the country enter the top hundred.

In 2022, Ukraine had better indicators of innovative products than innovative resources, as it was ranked 75th in terms of innovative resources, which was higher than last year (+1) but lower than that in 2020 (−4). Regarding innovative products, Ukraine ranked 48th, having lost 11 positions compared to 2021 and 2020 [35].

In general, according to the ratings, Ukraine ranks 4th among 36 countries with lower–middle-income economies below the average. However, among European countries, it ranks 34th among 39 European economies. According to the European Innovation Scoreboard (EIS), Ukraine is an Emerging Innovator, with a performance at 31% of the EU average. At the same time, since 2021, Ukraine has shown a significant increase in venture capital expenditures, PCT patent applications, and sales of innovative products [35].

The ratings of The Global Innovation Index [34] and the European Innovation Scoreboard [35] prove that human capital is the driver of Ukrainian innovative competitiveness. Its effective implementation is the main factor for obtaining a competitive advantage. Moreover, this research work goes a step further, focusing on the influence between ecosystems. In spite of this fact, authors such as Neumeyer et al. [36] have analyzed this issue. This particular focus on Ukraine adds novelty and solves some key questions about entrepreneurial development and gender approaches in significant crises such as the current war.

*2.2. Women's Entrepreneurship in Ukraine—A Theoretical Basis and Applied Aspects of Development*

After substantiating the influence of the entrepreneurial ecosystem, it is necessary to consider the connection between the institutional environment and the development of women's entrepreneurship, especially depending on whether the ecosystem falls under the category of barrier or non-barrier. The resources available in the entrepreneurial ecosystem will enable the activation of women's entrepreneurship. While the Index of Economic Freedom [32] highlights Ukraine's positive changes in this regard (before the start of the full-scale war with Russia), this country is still in the middle range of "mostly unfree" countries, which creates additional threats and challenges for the development of women's entrepreneurship.

Promoting the development of women's entrepreneurship is an important factor in economic growth in Ukraine. Researchers believe that problems regarding entrepreneurial sustainability and activity can also be exacerbated by gender-associated factors [37,38].

Concurrently, many women in organizations, who were previously mostly confined to the lower and middle management levels and, in the majority of firms, denied any opportunity to move into upper management [39,40], made the transition to private ownership [17]. That is why, taking their corporate experience and management style with them, they founded businesses at twice the rate of men and were equally successful [41]. According to the Global Entrepreneurship Monitor [15], in Europe, the total rates of initial entrepreneurial activity (TEA) tend to be low compared to those in other world regions but often reflect a high level of gender parity.

The results of international empirical comparative studies indicate that, in general, there is a clear statistical pattern that women are less involved than men in the creation of scientific and industrial knowledge [42,43]. Whilst women represent over 35% of all researchers in the higher education and government sectors of most European countries, this is not the case for the corporate sector [44].

The percentage of female researchers in the business sector is less than 25% in most countries. The authors of [44] investigated women's underrepresentation among holders of commercial patents.

Moreover, a growing body of evidence shows that organizations with a higher percentage of women in leadership roles outperform male-dominated companies. Unfortunately, however, women-owned companies do not receive the same level of financial backing as those founded by men [45,46].

LinkedIn data for 22 countries show that, in recent years, women have been establishing businesses at a slightly higher average rate than men. The share of women founders has doubled in the past five years, while the share of men founders has increased by 55%. In the same vein, the Global Entrepreneurship Monitor 2022 showed an encouraging figure for the first time, with the percentage of women involved in entrepreneurship for less than three and a half years exceeding that of men for the first time in 2021—5.6% compared to

5.4%, respectively. It should also be noted that this percentage has increased at all stages of the entrepreneurial process, although the indicators of women's representation are still much lower for high-tech companies [13]. For startups, the gender breakdown of their founders was 88% male-led compared to 12% female-led [47]. By contrast, the European average is 14% female startups [48], which is slightly higher than the global rate.

The Global Gender Gap Index measures the shares of women and men who occupy professional and technical roles, as well as senior official and manager roles. Women's share of senior and leadership roles has seen a steady global increase over the past five years (2017–2022). In 2022, global gender parity for this category reached 42.7%, the highest gender parity score yet.

Overall, the global share of women in leadership roles, as illustrated by these data, is 31%, although shares vary by industry. In 2022, only select industries had levels of female leadership near gender parity, such as non-governmental and membership organizations (47%), education (46%), and personal services and well-being (45%). At the other end of the range are energy (20%), manufacturing (19%), and infrastructure (16%) [47].

In 2021, women headed more than 33% of Ukrainian companies. The situation in Ukraine corresponds to the global trend: the share of female business leaders does not exceed 25–30%. For the most part, Ukrainian women entrepreneurs are sole owners of one enterprise. The lion's share of women's business is built around the sphere of services (44%) and trade (21%), while 10% of businesswomen are involved in education, 8% are involved in the creative industries, 5% have a business in HoReCa, 5% each are involved in the construction and IT industries, and 3% are in the agricultural sector.

The Ukrainian Investment and Trade Promotion Center (ITFC) [49] points out that female entrepreneurs subjectively and ambiguously assess the current state of entrepreneurship in Ukraine, with responses ranging from "continue their activity" to "completely or partially suspended their activity." Based on analytical studies and expert assessments [50], it can be concluded that, after the end of the war, women's entrepreneurship will become one of the key sources of development for the Ukrainian economy.

The experience of Ukrainian women entrepreneurs in overcoming economic crises is particularly valuable. In particular, the impact of the COVID-19 crisis on women's entrepreneurship was studied by Sörensson and Navid Ghannad [51] through the 4 Ds theoretical model: Dollars, Demand, Digitalization, and Distribution. They emphasized that entrepreneurs need to have a buffer of financial capital as a safety net when a crisis such as COVID-19 occurs. Another important component of overcoming the crisis was the digitalization of distribution processes, stimulating demand through social media. The importance of supplier/customer networks for women entrepreneurs during the crisis was confirmed by Mohapatra and Roy [52].

The outbreak of full-scale war in Ukraine in 2022 became the biggest challenge and the biggest stimulus for the development of women's entrepreneurship. In 2023, 56% of new private small businesses in Ukraine were started by women [53]. A study [54] conducted within the framework of the Good Governance Fund project "Revitalising the Business Climate in Ukraine", with the participation of the Ministry of Digital Transformation of Ukraine and the Ministry of Economy of Ukraine, confirms the data from the above theoretical studies. The main factor in the decision to become a woman entrepreneur is the availability of financial support and emotional support from one's family. Ukrainian women entrepreneurs consider the digitalization of business processes, the search for and systematization of information, and problems with human resources management to be important areas of development during crises. The in-depth interviews conducted in this research [54] showed that the top three needs for Ukrainian women entrepreneurs are as follows:

-   Financial and economic competence. The study participants pointed to a lack of skills in income forecasting, profitability assessment, and financial planning.
-   Digitalization of business processes. The participants indicated a lack of experience in digitalizing business processes and expressed hesitation about the feasibility of

implementing a CRM system in their business due to a lack of understanding of the benefits of such a system and its use in planning their business activities.

- Searching for and systematizing information. In the online survey, a significant proportion of the respondents (40%) indicated that lifelong learning is important for entrepreneurial competence.

The number of Ukrainian women aged 15–70 is 16.6 million, and only 7.8 million, or 47%, are part of the employed population. This indicates significant unused labor potential and, therefore, opportunities to increase the standard of living of the population [55].

According to the results of the analysis of the Unified State Register of Enterprises and Organizations of Ukraine (UEDRPOU), conducted within the framework of the project of the United Nations Development Program in Ukraine entitled "Strengthening business associations of small and medium-sized enterprises" [56], the overall ratio of men to women, as economic leaders in Ukraine, was 60:40. If we consider individual entrepreneurs and managers of legal entities who are included in the category of "economic leaders" separately, it turns out that gender inequality in this area is much more significant. Women represent only 30% of managers of legal entities.

According to the State Statistics Service of Ukraine, as of 1 November 2021, the share of businesses headed by women remains quite low. It is the highest among micro-enterprises but decreases with an increasing number of employees of the enterprise (Figure 1).

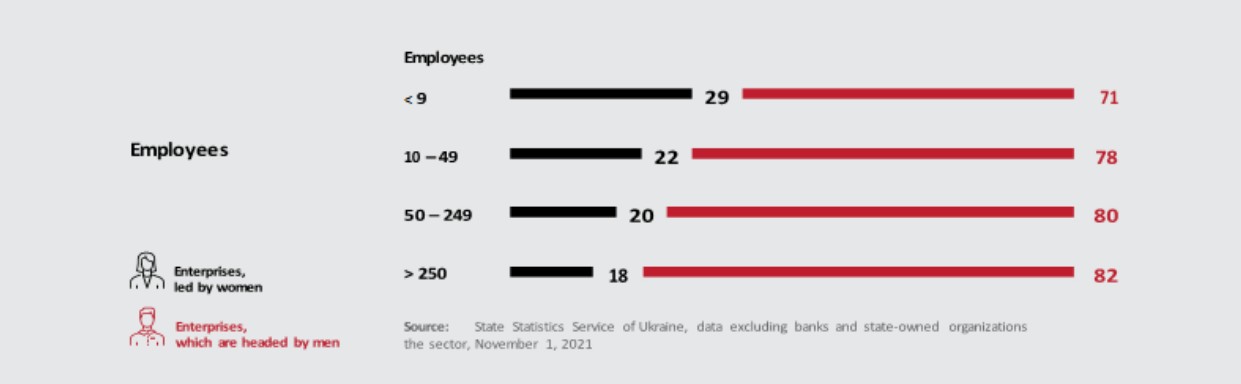

**Figure 1.** Share of enterprises in Ukraine headed by women/men (2021).

In the last few years, before the start of the war in Ukraine, the number of small businesses started by women was greater. In 2020, the number of small businesses started by women exceeded the number registered by men by 2%, and, in 2021, this gap widened to 2.5%, according to data from the Opendatabot study [53]. Data for 9 months of 2022 indicate that small businesses with a male founder are currently ahead of women's initiatives by more than 4.3%: 92,459 (more than 52%) new small businesses in Ukraine were founded by men. However, the situation in the first three months of 2023 has already changed; women in Ukraine started 189,776 small businesses, corresponding to 51% of the share of all registered individual entrepreneurs since the beginning of 2023 [53]. A third of new small businesses registered by women are in retail, almost twice as many as those registered by men in this sector. A total of 10,658 women entrepreneurs started their own businesses in the field of IT; this field ranked second place in the number of registrations in 2023 in Ukraine.

Based on monographic research, analyses of reports of global comparative systems [32–35], and analytical studies of the development of Ukrainian women's entrepreneurship during the war [50,57], the main hypotheses of the present study are put forward as follows:

**H1.** *The barrier/barrier-free character of women's entrepreneurship depends on the level of development of the existing institutional environment in which the entrepreneurial ecosystem functions.*

**H2.** *Ukrainian women entrepreneurs can be the basis for economic revival, according to their level of involvement in priority economic activities, depending on the following:*

**H2.1.** *The total number of women involved;*

**H2.2.** *The efficiency of each industry.*

It should be emphasized that the comprehensive analysis of factors affecting the functioning of the ecosystem of women's entrepreneurship during military operations is important. This approach contributes to an adequate understanding of the properties of the phenomena addressed, their possible transformations, and their consequences.

## 3. Methodology

A quantitative approach based on the main indices introduced above in the theoretical framework was used. Specifically, the Index of Economic Freedom [32], the EU Innovation Scoreboard [35], The Global Innovation Index [34], and Doing Business [33] were used to test the influence of the ecosystem on entrepreneurship (Hypothesis 1).

The analysis of international comparative indices and theoretical studies discussed in the literature review made it possible to formulate the first hypothesis. To test it at the first stage, we chose the method of correlation and regression analysis. The resulting factor (Y) is the net profit of small businesses. This is because this indicator is the goal of entrepreneurial activity. The factor (X) indicators were selected to characterize women's entrepreneurship in Ukraine in terms of official data (obtained from the State Statistics Service of Ukraine) and analytical studies of public authorities. Research on gender equality in Ukraine and the collection of analytical data from such research, including the list of indicators approved by the Order of the Cabinet of Ministers of Ukraine of 02 December 2020 No. 1517-r, "Issues of data collection for monitoring gender equality" (No. 1517-2020) [58], were used to collect data for monitoring gender equality. The data used in this study are presented in Appendix A. Based on these statistics (Appendix A), correlation and regression models were built, and the relationship between the selected indicators, characterizing the impact of various elements of the entrepreneurial ecosystem on women's business activities, was established. Correlation and regression equations allowed the determination of the indicators that have the greatest impact on the net profit of small businesses. A dependency analysis was conducted to determine the impact of the indicators.

The next step was to establish the relevant relationships between the indicators, which were previously substantiated by correlation and regression analyses. The indicators with the highest correlations were included in the simulation modeling of the impact of the indicators on the resulting characteristic.

Simulation modeling was carried out in the VENSIM environment "https://www.vensim.com/documentation/index.html (accessed on 14 April 2024)", designed for modeling and analyzing dynamic systems of various types [59].

Based on the results of simulation modeling and quantitative priority assessments, we divided all types of economic activity in which women entrepreneurs were most active (Hypothesis 2.2) into three groups. Thus, we formed optimistic, basic, and pessimistic forecasts for business structures created by women entrepreneurs (Hypothesis 2.1). For this purpose, the Harrington scale was used as a reference [60].

The use of simulation modeling allowed us to take into account the influence of indicators on the resulting feature (the net profit of small business enterprises). To substantiate the forecast of the development of women's entrepreneurship under the conditions of the existing entrepreneurial ecosystem, we substantiated the priorities of types of economic activity for female entrepreneurs.

In order to construct quantitative assessments of priority, the types of economic activity were chosen, for which the basic rate of growth/decline was used, based on quantitative indicators, as follows:

$$y_{ij} = \frac{x_{ij}(t_0)}{x_{ij}(T)} \tag{1}$$

where $x_{ij}(t_0)$ is the value of the $j$-th indicator for the $i$-th type of economic activity for the $t$-th year; $t_0$ is the base year selected 2011; and $T$ is the current year of observations 2020.

To determine the rate of development of small business enterprises according to the relevant types of economic activity, it is necessary to use the basic absolute growth equation, as follows:

$$y_{ij} = x_{ij}(T) - x_{ij}(t_0) \tag{2}$$

After the relevant calculations, the types of economic activity were ranked according to the priority of their development, taking into account the fact that these are small businesses whose managers are women. Then, the value of the priority function was calculated for each type of activity.

The priority function for types of economic activity according to the $j$-th indicator is as follows:

$$FP(rx_i, ry_i) = \frac{1}{m} \sum_{j=1}^{m} FP_j\left(rx_{ij}, ry_{ij}\right) \tag{3}$$

where $rx_i = (rx_{i1}, \dots, rx_{im})$ denotes the ranks of the $i$-th type of activity according to the last observation period, and $ry_i = (ry_{i1}, \dots, ry_{im})$ denotes the ranks of the $i$-th type of activity, according to the rates of growth/decline in indicators.

In accordance with the above statistical formulas, we developed ratings based on the growth rates of dynamic series, which were calculated in Excel for the years 2011–2020. These figures were released by the State Statistics Committee of Ukraine before the full-scale invasion. Then, based on the rating for each indicator, we determined the priority of the development of types of economic activity (according to Formula (3)). Types of economic activity for determining the priority areas of development of women's entrepreneurship were formulated in accordance with the legislative documents of Ukraine, specifically the methodological foundations and explanations as to the positions of the national classifier DK 009:2010 "Classification of types of economic activity" (KVED, 2010) [55]. In accordance with this regulatory document, the types of economic activity in modern Ukraine were derived.

We included all enterprises, arranged by type of economic activity, in the statistical sample. Then, on the basis of statistical data on the share of enterprises headed by women, we selected only those headed by women. In this way, the priority of development in general according to types of activities and the priority of development of enterprises headed by women were calculated. Based on this, we obtained the general priority of types of economic activity to develop a forecast for the development of women's entrepreneurship in Ukraine.

The results of the overall priority assessment were obtained according to a scale that was built on the basis of a probability distribution using a Harrington scale (Table 2). This scale will characterize the likely scenarios of the development of enterprises, according to the relevant type of economic activity, whose managers are women.

We chose these methods for research based on the fact that each change in an individual component leads to a change in the entire production and economic system, so a combination of two multivariate analyses, namely, regression analysis and simulation modeling, is effective.

In our case, a number of criteria were used, including the conformity of the type of economic activity with the sectoral and regional development program; the use of the dynamics of indicators that characterize the state of activity of small business entities; the number of employed populations by an enterprise of a given type, taking into account the number of women in the composition of the working population; managerial positions;

and small business owners. That is, not only was the current state of a business entity taken into account, but the prospects of its development in the future are also considered, with an orientation towards the development of women's entrepreneurship.

**Table 2.** The priority of the development of women's entrepreneurship arranged by type of economic activity.

| The Interval of Values of the Priority Function | Results |
|---|---|
| 1.00–0.8 | A very high level of indicators of the development of enterprises of selected types of economic activity, which should be supported and developed. |
| 0.8–0.63 | A high level of development indicators of enterprises engaged in selected types of economic activity that require support from the state and are desirable for development. |
| 0.63–0.37 | A satisfactory level of indicators of the development of enterprises, which are characterized by low growth rates and require moderate support. |
| 0.37–0.20 | A low level of development of entrepreneurship, which is characterized by low growth rates and needs significant support for its development. |
| 0.20–0.00 | A very low level of development of entrepreneurship in the selected sector of the economy, which is characterized by a lack of growth and requires an assessment of the feasibility of its support and development. |

## 4. Results

Based on the construction of correlation–regression models, the dependence between the selected indicators that characterize the impact of the macro environment on the activities of women in business was established. A total of 24 indicators were investigated, but 18 were used in the simulation, in which the coefficient of determination, $R^2$, had a value greater than 0.5, which characterizes the greater dependence of one value on another. The general calculation of the coefficient of determination has the following mathematical form:

$$R^2 = 1 - \frac{D[yx]}{D[y]} = 1 - \frac{\delta^2}{\delta_y^2} \tag{4}$$

where $D[y] = \delta_y^2$ is the variance of the random variable y, and $D[yx]$ is the conditional (according to factors of $x$) variance of the dependent variable (model error variance).

The following indicators were chosen for modeling (Table 3).

To carry out the modeling process, indicators were selected, taking into account the gender distribution according to the relevant indicators. As a result of modeling, it was established that the main indicator of the influence on the development of a small enterprise is the volume of industrial products (goods, services) and added value. Among small enterprises and individual entrepreneurs, a key factor is the share of women among enterprise managers and entrepreneurs.

In order to establish the relevant dependencies of the indicators substantiated by correlation–regression analysis ($X_1$–$X_{24}$), simulation modeling of the influence of such indicators (net profit of small business enterprises, thousand hryvnias) was carried out.

The simulation model was built on the basis of establishing a direct feedback relationship between the selected indicators, which formed 32 contours, containing between two and seven variable models. Indicators $X_{14}$–$X_{16}$, $X_{20}$, $X_{21}$, and $X_{23}$ were not included in the simulation model due to low $R^2$ values.

When simulating the interrelationships of the components of the development of small business enterprises among women under the influence of entrepreneurship ecosystem

factors, various streams are included that are aimed at the innovative development of the state economy and the promotion of gender equality in society.

**Table 3.** Indicators for modeling.

| | Indicator | Value ($R^2$) |
|---|---|---|
| y | Net profit of small business enterprises (thousand hryvnias) | |
| $X_1$ | Volume of industrial products (goods, services) sold by small enterprises (million hryvnias) | 0.87 |
| $X_2$ | Number of employees, women in small business enterprises (thousands of people) | 0.62 |
| $X_3$ | Costs for innovation (million hryvnias) | 0.53 |
| $X_4$ | Share of women among managers of enterprises and entrepreneurs according to type and size of settlement | 0.81 |
| $X_5$ | Added value based on production costs of business entities of individual entrepreneurs (thousand hryvnias) | 0.84 |
| $X_6$ | Number of operating large enterprises (units) | 0.61 |
| $X_7$ | Number of operating medium-sized enterprises (units) | 0.50 |
| $X_8$ | Number of operating small enterprises (units) | 0.62 |
| $X_9$ | Number of active information-and-communication technology enterprises (units) | 0.70 |
| $X_{10}$ | Number of active enterprises of information and communication technology in production (units) | 0.58 |
| $X_{11}$ | Number of active enterprises of information and communication technologies in service (units) | 0.73 |
| $X_{12}$ | Investment in fixed assets (mln hryvnias) | 0.80 |
| $X_{13}$ | Number of active production enterprises using medium–high-level technologies (units) | 0.58 |
| $X_{14}$ | Number of operating enterprises for production using medium–low-level technologies (units) | 0.04 |
| $X_{15}$ | Number of operating enterprises for production using low-level technologies (units) | 0.01 |
| $X_{16}$ | Number of operating enterprises in the information sector (units) | 0.35 |
| $X_{17}$ | Number of operating enterprises from services using high-level technologies (units) | 0.5 |
| $X_{18}$ | Number of operating enterprises from intellectually saturated market services (units) | 0.58 |
| $X_{19}$ | Number of operating enterprises providing services related to the use of computer equipment (units) | 0.79 |
| $X_{20}$ | Number of operating enterprises from creative industries (units) | 0.01 |
| $X_{21}$ | Business incubators (units) | 0.19 |
| $X_{22}$ | Volumes of invested venture capital (million dollars) | 0.75 |
| $X_{23}$ | Number of innovation centers (units) | 0.02 |
| $X_{24}$ | Number of companies started by women (units) | 0.54 |

In the model, feedback contours are defined by green lines, i.e., those relationships that have both a direct and an inverse relationship with the selected indicators (Figure 2). The contours of the simulation model have the following structure:

Volume of industrial products (goods, services) sold by small enterprises (million hryvnias) → added value based on production costs of business entities of individual entrepreneurs (thousand hryvnias) → net profit of small business enterprises (thousand hryvnias);

Costs for innovation (million hryvnias) → number of operating large enterprises (units) → number of active information and communication technology enterprises (units) → volume of industrial products (goods and services) sold by small enterprises (million hryvnias);

Investment in fixed assets (mln hryvnias) → volumes of invested venture capital (million dollars) → number of operating small enterprises (units) → number of compa-

nies started by women (units) → share of women among managers of enterprises and entrepreneurs (according to type and size of settlement).

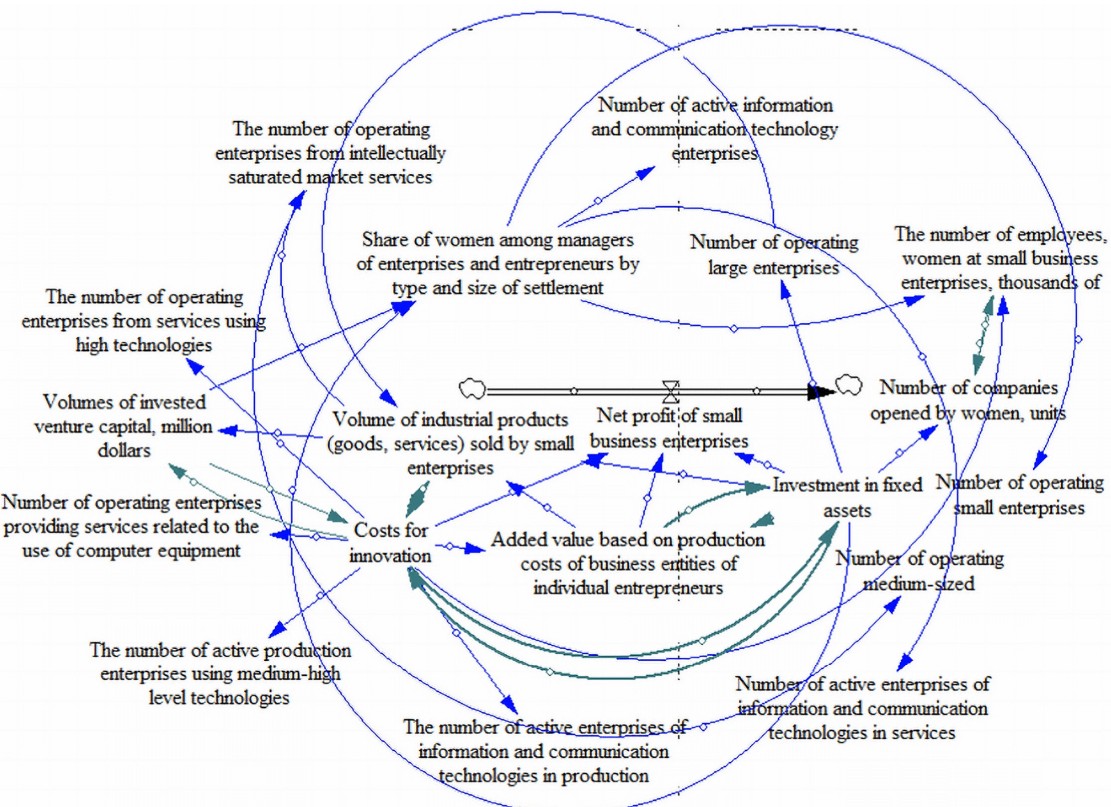

**Figure 2.** Simulation modeling of the influence of environmental factors on the development of enterprises (headed by women).

According to the contours of simulation modeling of the influence of environmental factors on the development of enterprises (headed by women), the main indicators of influence on the development of women's entrepreneurship are the volume of industrial products (goods, services) sold by small enterprises, the costs of innovation, and investment in fixed assets.

The use of simulation modeling allows for the consideration of the influence of the relevant indicators on the resulting feature. In our case, the net profit of small enterprises was chosen as the resulting indicator since profit is the main goal of the creation and operation of enterprises of any type. Depending on the industry, the profit and operating conditions of enterprises vary significantly.

As can be seen from the analytical data, the majority of women managers and business owners are part of small enterprises and within the sphere of economic activity, which are more related to trade, education, and the provision of services. In order to establish a forecast for the development of small businesses whose managers are women, it is advisable to prioritize the types of economic activities for their further development, precisely on the condition that the enterprises are managed by women.

Based on the results of the correlation–regression analysis conducted in SPSS (Appendix B), the results of simulation modeling, and the ranking of the priority of the development of types of economic activity of small business entities managed by women, which we calculated on the basis of growth rates per industry, we formed a forecast for the years 2024–2026.

According to the corresponding results, the forecast was made for three years (2024–2026) since, during the period of military operations and in the absence of the necessary analytical tools, a rather high error in forecasts is possible for the longer term. The results

of calculating the priority of support for each type of economic activity according to the number of small business entities are shown in Table 4.

**Table 4.** Grouping of types of economic activity of enterprises headed by women according to the priority of development and development scenarios. The types of economic activity are given according to NACE-2010.

| Types of Economic Activity | Sorting by Priority of Types of Economic Activity Involving Women Acting as Managers | | General Forecast of the Development of Small Businesses among Women According to Industry Type |
|---|---|---|---|
| Wholesale and retail trade, repair of motor vehicles and motorcycles | 0.89 | Very high level by priority | Optimistic development scenario |
| Provision of other types of services | 0.78 | High level by priority | |
| Professional scientific and technical activity | 0.78 | | |
| Real estate transactions | 0.78 | | |
| Activities in the field of administrative and auxiliary services | 0.77 | | |
| Temporary accommodation and catering | 0.73 | | |
| Education | 0.72 | | |
| Information and telecommunications | 0.7 | | |
| Wholesale and retail trade, repair of motor vehicles and motorcycles | 0.69 | | |
| Provision of other types of services | 0.68 | | |
| Professional scientific and technical activity | 0.68 | | |
| Real estate transactions | 0.68 | | |
| Activities in the field of administrative and auxiliary services | 0.64 | | |
| Temporary accommodation and catering | 0.62 | Satisfactory level by priority | Basic development scenario |
| Water supply, sewage, and waste management | 0.44 | | |
| Supply of electricity, gas, steam, and air conditioning | 0.4 | | |
| Mining and quarrying | 0.39 | | |
| Activities of households | 0.34 | Low level by priority | Pessimistic development scenario |
| Activities of exterritorial authorities | 0.33 | | |
| Processing industry | 0.29 | | |
| Construction | 0.2 | | |

As evidenced by the data in Table 4, according to the number of small business entities arranged by type of economic activity, those that need support are those that have a high-level priority value of 0.63–0.8 or 0.8–1.0; these businesses characterize an optimistic scenario of development, as they have high indicators of their development and are important for the state in terms of supporting its socio-economic situation. The key assumption for the optimistic scenario is that there will be no dramatic changes in the share of women entrepreneurs per industry in the short term. Accordingly, in the short term, the distribution of industries in which women entrepreneurs are an important driver of development will remain as we have predicted.

The basic scenario includes those types of economic activity that are in the range of 0.37–0.63, as they are not characterized by high growth rates and require moderate support for their development.

Those types of economic activity that are in the 0.37–0.20 and 0.2–0.00 ranges are characterized by a low and very low level of priority, a lack of development, and a requirement for significant support for their activities and can be assigned to the pessimistic development scenario. For such types of economic activity, if they are strategically important for the state, it is advisable to consider the feasibility of their development or to introduce new forms and methods of conducting business to increase their priority and economic feasibility. This category includes the construction and processing industries, because this area of business activity was traditionally "not for women". However, after the end of the war, the construction sector is potentially a priority. This, in turn, will allow for the additional development of related industries, such as wholesale and retail trade, and other types of services. In other words, this is an additional confirmation of the possibility of implementing an optimistic scenario for the development of women's entrepreneurship in Ukraine.

As a result, three scenarios of the development of women's entrepreneurship were formed, based on types of economic activity, which are priorities for the development of entrepreneurship after the war (optimistic, Figure 3; basic, Figure 4; and pessimistic, Figure 5).

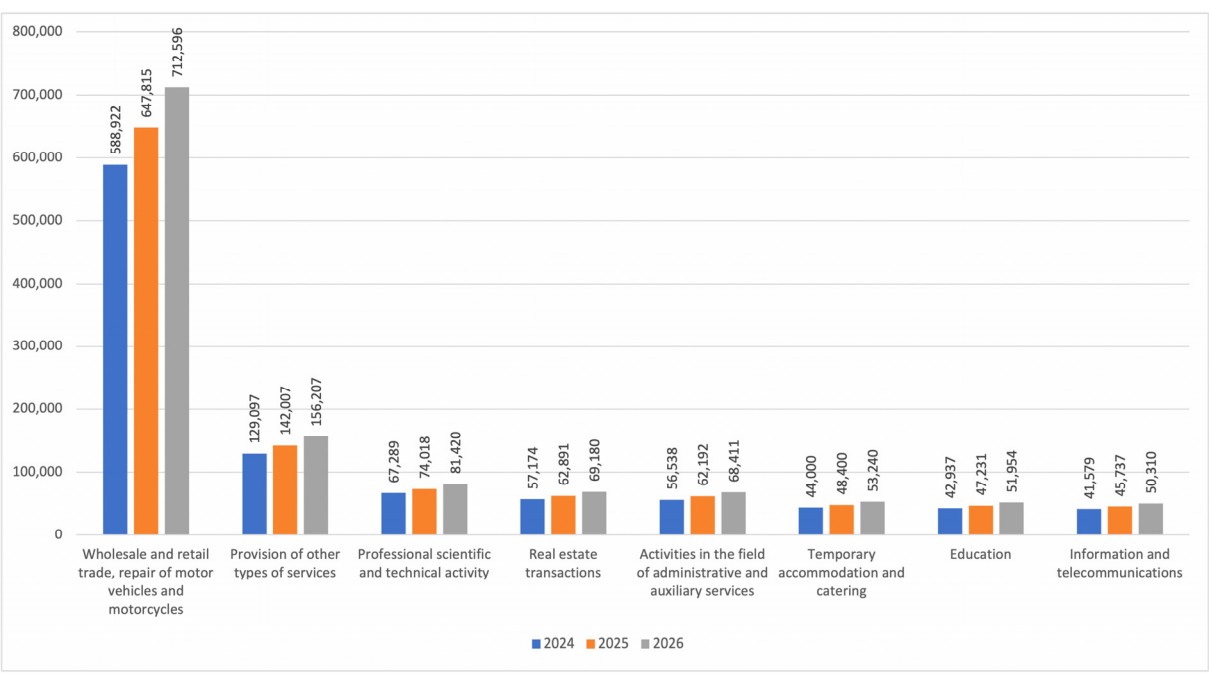

**Figure 3.** Forecast according to the optimistic scenario of small business development according to type of economic activity (with women as heads of units).

Based on the conducted research, three types of development scenarios of women's entrepreneurship in Ukraine were established. Accordingly, the highest-priority types of economic activity for women entrepreneurs are wholesale and retail trade, the provision of other types of services, professional scientific and technical activity, and real estate transactions. It is precisely these areas that will be developed primarily thanks to women.

The optimistic scenario for the development of women's entrepreneurship for the period up to 2026 is explained by the choice of economic activities, in which there has been a significant share of women entrepreneurs for a long time. They have maintained these industries during the war and will continue to develop them after the end of hostilities. Post-war reconstruction should give an additional impetus to the development of women's entrepreneurship. According to a survey conducted in February 2023 by the NGO Centre for the Development of Corporate Social Responsibility, every third woman is engaged in the service sector: trade, fashion, and beauty [57]. However, this research work does

not determine the potential of emergent sectors for women as technological companies or entrepreneurs represented by startups [18,26].

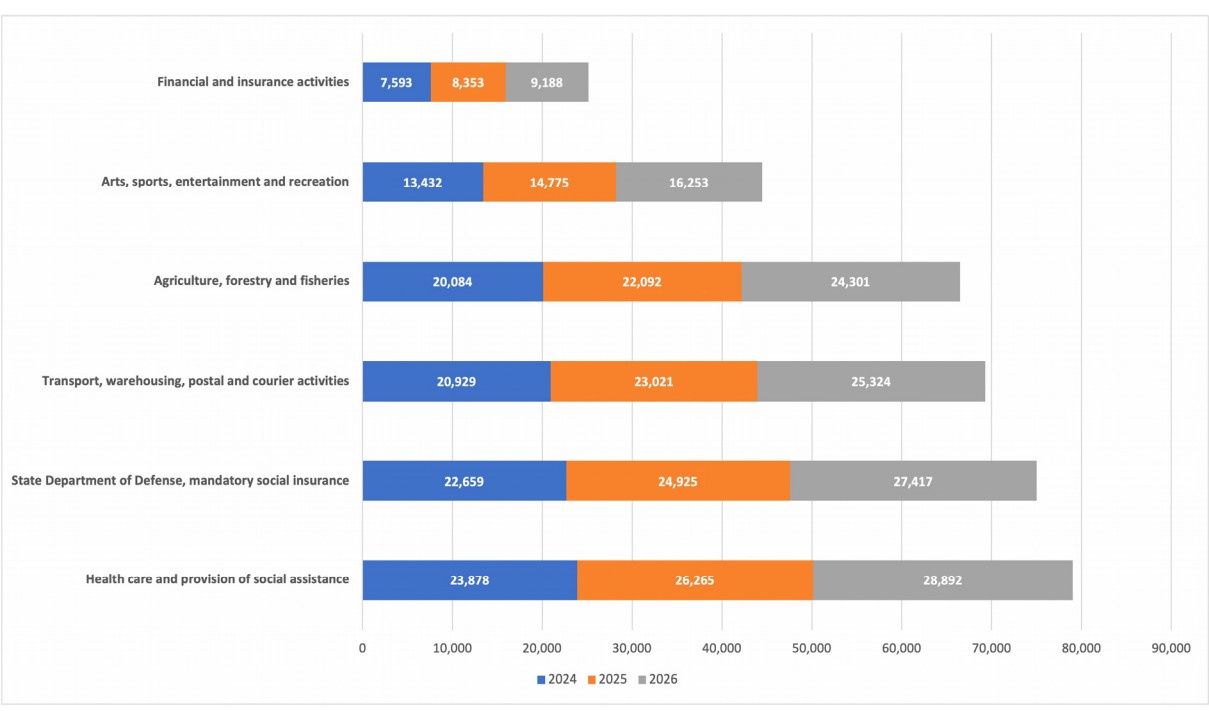

**Figure 4.** Forecast according to the basic scenario of small business development according to type of economic activity (with women as heads of units).

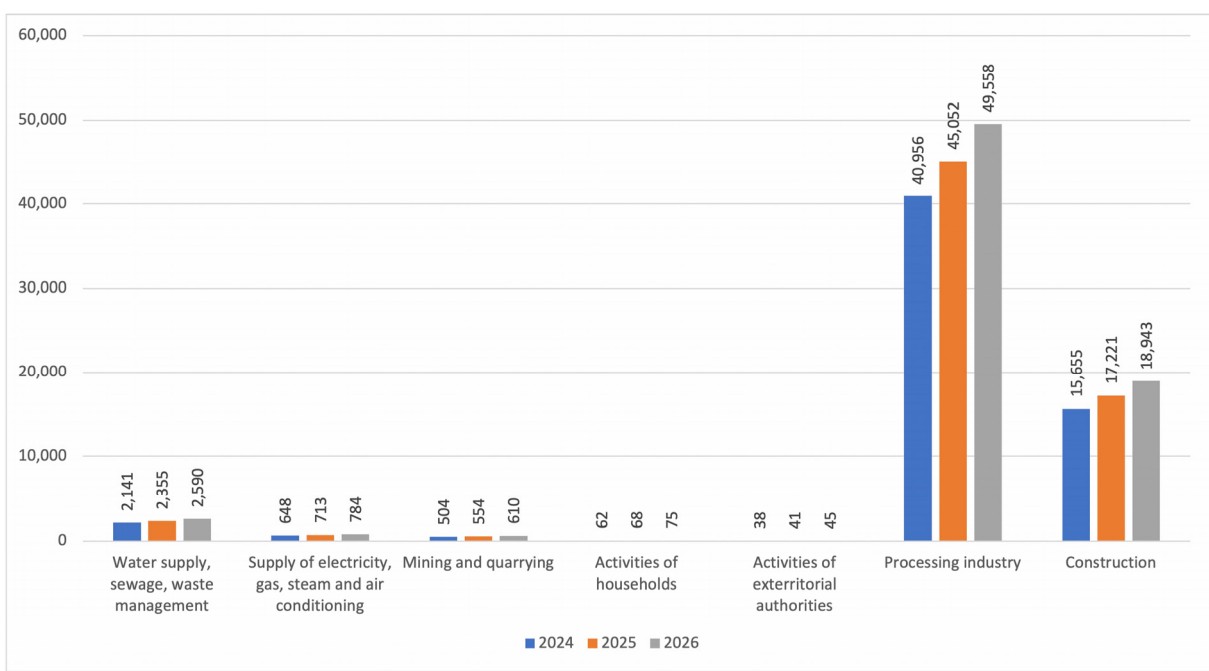

**Figure 5.** Forecast according to the pessimistic scenario of small business development according to type of economic activity (women as heads of units).

The basic scenario of small business development enables the gradual development of business without significant losses and investments, which contributes to a business's development in accordance with the life cycle of production. The pessimistic scenario of the development of women's entrepreneurship includes types of economic activity in

which women's work is at a low level due to the complexity of the work and requires large investment funds. These types of activities include industry, construction, and the fields of energy and public administration.

## 5. Discussion and Conclusions

According to the distribution of priority types of economic activity in relation to leadership among women, we forecast the development of those types of economic activity that, according to our calculations, have the highest level of priority. However, among the industries, there are those that have a low level of priority from the point of view of their economic benefit but that are strategically important for the development of the state and regions, and their support at a certain level is necessary. Small and medium-sized enterprises in the trade and services sector should receive the greatest support for development in women's entrepreneurship in the post-war period, as the restoration and development of large industrial enterprises require significant funds and time. This is consistent with the female representation in the economics of entrepreneurship amply demonstrated in previous work, regardless of whether there is a situation as complex as a war, a specific contribution and novelty of this research work. On the basis of our research, a forecast of the development of women's entrepreneurship in Ukraine according to economic activity type was made, which can become one of the benchmarks for the post-war recovery of Ukraine.

According to the hypotheses of the study, the following were established:

1. The ecosystem of Ukrainian entrepreneurship in the pre-war period, according to the investigated indices [32,34,56], showed an increase in the availability and ease of doing business. These indices were used to test the impact of ecosystems on entrepreneurship (Hypothesis 1). Hypothesis 1 was proven on the basis of monographic studies and expert assessments, which confirmed that Ukrainian women entrepreneurs face greater challenges in their activities than men (ITFC) (2022), even though in Ukraine, even under conditions of war, the conditions for the development of women's entrepreneurship remain [50]. The simulation model showed that the main indicators of the impact on the development of women's entrepreneurship are the volume of industrial products (goods, services) sold by small enterprises, the costs of innovation, and investments in fixed assets.

2. Women's entrepreneurship, taking into account economic and demographic factors, will become one of the most effective components of the post-war revival of Ukraine. On the basis of forecasting, the following priority types of economic activity for women entrepreneurs were determined: wholesale and retail trade, the provision of other types of services, professional scientific and technical activities, and transactions involving real estate.

In particular, in the field of wholesale and retail trade, the number of women entrepreneurs is predicted to increase from 588,922 in 2024 to 712,596 in 2026 (by 21%). While their number was 564,381 in 2020, the projected growth in the number of women entrepreneurs in this field by 2024 amounts to 4,3 percent. We assume that, after 2024, the post-war revival of the country will begin. Consequently, Hypotheses 2.1 and 2.2 are confirmed.

This research proves the importance and value of women's entrepreneurship, which is important both during the war and for the post-war reconstruction of the Ukrainian economy. However, women's entrepreneurship continues to face additional challenges and obstacles. As researchers highlight [10,11], this is a "different" type of female entrepreneurship. That is, until now, especially in countries with low and medium levels of economic development, women's entrepreneurship has faced additional obstacles and discrimination, as in Ukraine. To some extent, the attributions of sectors of activity derived from the model are also a consequence of those barriers that remain in place in Ukraine and that perpetuate gender differences in representation in some sectors. This is confirmed by studies [21] that analyzed the development of female entrepreneurship in Central and Eastern Europe and

concluded that patriarchal and neoconservative concepts of gender have been preserved in these countries.

### 5.1. Practical Implications

The practical implications of this study can be used by public authorities in planning the development of entrepreneurship in Ukraine after the end of the war. The results of entrepreneurial activity, namely, the number of newly registered women entrepreneurs over the past two years, confirm that women have become more proactive [53]. Therefore, it would be advisable to consider the development of the industries included in the optimistic scenario to create more favorable conditions for the development of women's entrepreneurship. Conducted studies [61] prove that the gender of the entrepreneur is not of key importance for economic growth. This will allow for a more rational distribution of labor resources in the context of the post-war reconstruction of Ukraine. Finally, women's contributions to the economy can be highlighted as a key player, even in particularly complex situations such as a military conflict. Thus, these conclusions could inspire the determination of inclusive policies for rebuilding the economy in conflict-ridden countries.

### 5.2. Limitations and Future Research Lines

This study has certain limitations that need to be addressed in future research. First, this research was conducted during the war, but the conclusions are focused on the processes of the post-war revival of Ukraine. The duration of the war is an unpredictable factor. The longer the war in Ukraine lasts, the more difficult it will be to develop women's entrepreneurship. This is primarily due to population migration. According to data presented in [62], as of the end of January 2024, 4.9 million Ukrainians are abroad because of the war. The vast majority of refugees are women and children.

Second, we considered the state and prospects of women's entrepreneurship in Ukraine without comparing them to those of other countries that have already had experience of post-war revival. Therefore, in further research, it would be advisable to study the experience of women's entrepreneurship development in countries that have experienced periods of war and post-war revival in recent decades (e.g., former Yugoslavia or Israel).

In the context of military operations, the socio-economic, spiritual, and cultural transformation of Ukrainian society is taking place, and the role of women in society is changing. Accordingly, it is advisable to develop our research in the context of the impact of and interrelationships between the quality of the institutional environment, the quality of entrepreneurial ecosystems, the level of development of women's entrepreneurship, and the overall well-being of the country.

In any case, making forecasts and conducting predictive analyses regarding the capacity to reorganize a country's productive model, especially focusing on how the female-led economy can be strategic, will encourage researchers to treat this limitation of this study as a dynamic topic for studies on entrepreneurship and sustainability in the future.

**Author Contributions:** Conceptualization: T.S., P.P.I.-S., C.J.-M., E.F.-D. and C.d.l.H.-P.; methodology: T.S., P.P.I.-S., C.J.-M., E.F.-D. and C.d.l.H.-P.; software: T.S., P.P.I.-S., C.J.-M., E.F.-D. and C.d.l.H.-P.; validation: T.S., P.P.I.-S., C.J.-M., E.F.-D. and C.d.l.H.-P.; formal analysis: T.S., P.P.I.-S., C.J.-M., E.F.-D. and C.d.l.H.-P.; investigation: T.S., P.P.I.-S., C.J.-M., E.F.-D. and C.d.l.H.-P.; resources: T.S., P.P.I.-S., C.J.-M., E.F.-D. and C.d.l.H.-P.; data curation: T.S., P.P.I.-S., C.J.-M., E.F.-D. and C.d.l.H.-P.; writing—original draft preparation: T.S., P.P.I.-S., C.J.-M., E.F.-D. and C.d.l.H.-P.; writing—review and editing: T.S., P.P.I.-S., C.J.-M., E.F.-D. and C.d.l.H.-P.; visualization: T.S., P.P.I.-S., C.J.-M., E.F.-D. and C.d.l.H.-P.; supervision: T.S., P.P.I.-S., C.J.-M., E.F.-D. and C.d.l.H.-P.; project administration: T.S., P.P.I.-S., C.J.-M., E.F.-D. and C.d.l.H.-P.; funding acquisition: T.S., P.P.I.-S., C.J.-M., E.F.-D. and C.d.l.H.-P. All authors have read and agreed to the published version of the manuscript.

**Funding:** This document has been funded by Proyectos de Generación de Conocimiento 2022, Ministry of Science and Innovation, State Research Agency (FEDER, EU). Grant number: PID2022-139037OB-I00/AEI/10.13039/501100011033/FEDER, UE; and Funding for Open Access Charge: Universidad de Málaga/CBUA.

**Institutional Review Board Statement:** Not applicable.

**Informed Consent Statement:** Not applicable.

**Data Availability Statement:** Data is contained within the article.

**Conflicts of Interest:** The authors declare no conflicts of interest.

## Appendix A

Data for Calculating the Correlation–Regression Analyses According to the State Statistics Service of Ukraine.

| Years | Y | $X_1$ | $X_2$ | $X_3$ | $X_4$ | $X_5$ | $X_6$ | $X_7$ | $X_8$ | $X_9$ |
|---|---|---|---|---|---|---|---|---|---|---|
| 2011 | 32,518.8 | 607.8 | 6852.4 | 14,333.9 | 12 | 61,266,261.9 | 659 | 20,753 | 354,283 | 9882 |
| 2012 | 35,296.2 | 672.7 | 6821.3 | 11,480.6 | 14 | 61,266,261.9 | 698 | 20,189 | 344,048 | 10,225 |
| 2013 | 35,748.2 | 670.3 | 6932.4 | 9562.6 | 16 | 61,266,261.9 | 659 | 18,859 | 373,809 | 11,562 |
| 2014 | 45,236.6 | 705.0 | 7123.6 | 7695.9 | 21 | 59,505,414.6 | 497 | 15,906 | 324,598 | 10,534 |
| 2015 | 89,390.4 | 937.1 | 7872.40 | 13,813.7 | 23 | 67,021,737.6 | 423 | 15,203 | 327,814 | 10,998 |
| 2016 | 99,298.7 | 1177.4 | 7827.40 | 23,229 | 24 | 102,918,629.8 | 383 | 14,832 | 291,154 | 9979 |
| 2017 | 107,934.7 | 1482.0 | 7771.20 | 9117.5 | 26 | 157,792,645.5 | 399 | 14,937 | 322,920 | 11,271 |
| 2018 | 127,658.9 | 1766.2 | 7910.70 | 12,180.1 | 28 | 200,075,980.3 | 446 | 16,057 | 339,374 | 12,291 |
| 2019 | 162,563.0 | 1839.9 | 7923.10 | 14,220.9 | 32 | 229,340,446.7 | 518 | 17,751 | 362,328 | 13,521 |
| 2020 | 142,204.9 | 2064.1 | 7605.80 | 14,406.7 | 49 | 257,624,371.3 | 512 | 17,602 | 355,708 | 13,829 |
| 2021 | 228,823.7 | 2103.3 | 7832.1 | 14,501.3 | 50 | 263,265,413.1 | 514 | 17,932 | 361,236 | 14,040 |

| Years | $X_{10}$ | $X_{11}$ | $X_{12}$ | $X_{13}$ | $X_{14}$ | $X_{15}$ | $X_{16}$ | $X_{17}$ | $X_{18}$ | $X_{19}$ |
|---|---|---|---|---|---|---|---|---|---|---|
| 2011 | 466 | 9416 | 1297 | 5241 | 15,954 | 18,221 | 7260 | 16,694 | 70,717 | 5148 |
| 2012 | 397 | 9828 | 1122 | 5038 | 14,327 | 16,280 | 6404 | 15,652 | 72,318 | 5220 |
| 2013 | 438 | 11,124 | 1210 | 5680 | 16,334 | 18,175 | 6861 | 17,100 | 79,124 | 6070 |
| 2014 | 375 | 10,159 | 1107 | 5019 | 13,739 | 16,013 | 5992 | 15,189 | 70,235 | 5633 |
| 2015 | 354 | 10,644 | 1068 | 5151 | 13,744 | 16,037 | 5864 | 15,397 | 72,139 | 5961 |
| 2016 | 286 | 9693 | 916 | 4895 | 12,468 | 14,156 | 4866 | 13,244 | 64,969 | 5350 |
| 2017 | 285 | 10,986 | 961 | 5337 | 13,485 | 15,414 | 5272 | 14,806 | 72,331 | 6264 |
| 2018 | 293 | 11,998 | 1001 | 5647 | 14,036 | 16,178 | 5495 | 15,859 | 77,135 | 7003 |
| 2019 | 320 | 13,201 | 1040 | 5931 | 14,799 | 17,005 | 5848 | 17,173 | 83,254 | 8063 |
| 2020 | 350 | 13,479 | 1082 | 6063 | 14,877 | 17,035 | 5613 | 17,232 | 80,275 | 8433 |
| 2021 | 365 | 13,675 | 1162 | 6342 | 15,219 | 17,357 | 5605 | 17,507 | 79,315 | 8822 |

| Years | $X_{20}$ | $X_{21}$ | $X_{22}$ | $X_{23}$ | $X_{24}$ |
|---|---|---|---|---|---|
| 2011 | 19,368 | 25 | 24 | 46 | 4100 |
| 2012 | 18,172 | 32 | 59 | 51 | 5320 |
| 2013 | 20,061 | 71 | 89 | 48 | 7652 |
| 2014 | 17,635 | 80 | 39 | 53 | 8452 |
| 2015 | 17,652 | 53 | 132 | 69 | 10,560 |
| 2016 | 14,982 | 55 | 88 | 68 | 12,300 |
| 2017 | 16,467 | 62 | 259 | 79 | 16,120 |
| 2018 | 17,474 | 48 | 337 | 81 | 10,320 |
| 2019 | 19,162 | 36 | 510 | 76 | 20,000 |
| 2020 | 18,932 | 28 | 580 | 42 | 18,230 |
| 2021 | 19,063 | 24 | 480 | 16 | 9450 |

## Appendix B

The Results of the Correlation–Regression Analyses of the Initial Indicators.

| | $X_1$ | $X_2$ | $X_3$ | $X_4$ | $X_5$ | $X_6$ | $X_7$ | $X_8$ | $X_9$ | $X_{10}$ | $X_{11}$ | $X_{12}$ |
|---|---|---|---|---|---|---|---|---|---|---|---|---|
| $R^2$ | 0.87 | 0.62 | 0.53 | 0.81 | 0.84 | 0.61 | 0.50 | 0.62 | 0.70 | 0.58 | 0.73 | 0.80 |
| F | 64,75 | 14.82 | 0.89 | 38.68 | 48.05 | 2.46 | 0.70 | 0.21 | 21.92 | 3.53 | 24,17 | 1.01 |
| | $X_{13}$ | $X_{14}$ | $X_{15}$ | $X_{16}$ | $X_{17}$ | $X_{18}$ | $X_{19}$ | $X_{20}$ | $X_{21}$ | $X_{22}$ | $X_{23}$ | $X_{24}$ |
| $R^2$ | 0.58 | 0.00 | 0.00 | 0.35 | 0.50 | 0.58 | 0.79 | 0.00 | 0.19 | 0.75 | 0.02 | 0.54 |
| F | 12.55 | 0.03 | 0.01 | 4.99 | 1.05 | 3.61 | 35.15 | 0.00 | 2.12 | 28.14 | 0.27 | 4.76 |

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
