# Peer review of "Ukrainian Women’s Entrepreneurship and Business Ecosystem during the War: New Challenges for Development"

_sustainability, doi:10.3390/su16093829_

Round 1

Reviewer 1 Report

Comments and Suggestions for Authors

Thank you for the opportunity to review the manuscript "Ukrainian women's entrepreneurship and business ecosystem during the war: new challenges for development". After carefully assessing the paper, I believe it merits publication, but would benefit from some minor revisions. The authors should be able to address these points within a few weeks.

The paper provides a timely and valuable analysis of the state of women's entrepreneurship in Ukraine and how the entrepreneurial ecosystem influences its development, especially in the challenging context of war. The use of correlation-regression analysis and simulation modeling to forecast scenarios for women's entrepreneurship is a key strength.

Some suggestions for improvement:

1. The literature review could be strengthened by discussing a few more recent studies on women's entrepreneurship, especially any that look at the impact of crisis/conflict situations. This would help situate the current work within the latest scholarship.

2. The methodology section would benefit from a bit more detail on the specific regression models used and model diagnostics to give readers confidence in the robustness of the results. 

3. When presenting the three forecast scenarios, it would be useful to discuss the key assumptions underlying each one and the degree of uncertainty around the projections. What factors could cause deviations from these trajectories?

4. The practical implications of the findings for policymakers and entrepreneurial ecosystem stakeholders could be fleshed out more in the discussion/conclusion section. What specific actions could help realize the optimistic scenario?

5. While the writing is generally clear, the manuscript would benefit from an additional round of proofreading and polishing to fix minor grammatical issues and improve flow in a few places.

6. Using an appendix for some of the detailed data tables is an efficient approach, but a couple key summary tables in the main text could help readers interpret the results more easily.

Overall, this is a well-designed study on an important and understudied topic. With some minor revisions to enhance clarity and strengthen the linkages to prior work, it will make a solid contribution to the entrepreneurship literature. 

Comments on the Quality of English Language

no major issues

Author Response

We thank the editors and the reviewers for their interest in our work and for helpful comments that will greatly improve the manuscript. The changes suggested has been helpful and they have improved the final manuscript. The Reviewers have brought up some good points and we appreciate the opportunity to clarify the way to express some questions, and introduce details which clarify some aspects in our research.

We have tried to do our best to respond to the points raised. In accordance with your suggestions, we have undertaken a revision of the manuscript and they are in green colour.  Do not hesitate to contact us in the case that any further details or improvements are needed. We will be pleased to respond to them promptly.

Yours faithfully,

The authors

REVIEWER 1

The Reviewer has brought up some good points and we appreciate the opportunity to clarify the way to express some questions, and introduce details which clarify some aspects in our research. We thank the reviewer for his comments and suggestions to improve our research.

  1. 1. The literature review could be strengthened by discussing a few more recent studies on women's entrepreneurship, especially any that look at the impact of crisis/conflict situations. This would help situate the current work within the latest scholarship.

In the literature review, we have added an overview of several theoretical researches that were published in 2024. We also analysed a report that was published with the support of the Ukrainian government in January 2024. It systematised the main needs and problems of Ukrainian women entrepreneurs during the war.

In the text of the article, these are lines 203-232, which are marked in red colour.

  1. The methodology section would benefit from a bit more detail on the specific regression models used and model diagnostics to give readers confidence in the robustness of the results. 

The  correlation and regression models illustrate the cause-and-effect relationships between the indicators that characterise the impact of the entrepreneurial ecosystem on women's business activities. For further use in the simulation model, we used only 18 out of 24 indicators with a determination coefficient R2 accurate statistically, always provided that they started from a significant correlation

On the basis of the presented mathematical model, a simulation model was developed, the development of which was preceded by the study of software capabilities to solve the problem.

The simulation model is implemented in the VenSim software product, in which the solution is presented in the form of levels corresponding to the modules and blocks presented in the conceptual model. More detailed information is added in lines 281-304, which are highlighted in red colour.

  1. When presenting the three forecast scenarios, it would be useful to discuss the key assumptions underlying each one and the degree of uncertainty around the projections. What factors could cause deviations from these trajectories?

Thank you for your comments. We have tried to clarify these discussion points more clearly by adding explanations on lines 456-460 and 470-476

  1. The practical implications of the findings for policymakers and entrepreneurial ecosystem stakeholders could be fleshed out more in the discussion/conclusion section. What specific actions could help realize the optimistic scenario?

Practical implications are added in the lines 560-571

  1. While the writing is generally clear, the manuscript would benefit from an additional round of proofreading and polishing to fix minor grammatical issues and improve flow in a few places.

Thank you for your warning. We have reviewed with a prof-editing by MPDI the current version of the manuscript.

  1. Using an appendix for some of the detailed data tables is an efficient approach, but a couple key summary tables in the main text could help readers interpret the results more easily.

For a more detailed understanding, Appendix 2 - The results of the correlation-regression analysis of the initial indicators, which summarises the results of the correlation-regression analysis, has been added. Line 620.

Reviewer 2 Report

Comments and Suggestions for Authors

Dear authors,

congratulations for choosing an interesting and challenging topic of your research. However, there are some  issues that need to be solved before the paper can be accepted:

-first of all, the paper needs language editing (grammar, syntax, too long and difficult to understand sentences)

-over half of section 2.2.  talks about global issues in women entrepreneurship, while, according to the title is should concentrate on situation in Ukraine

- regarding simulation model it would be very helpful to add some more detailed explanations which you refer to later in the conclusion

-it is not very clear how you confirm H1 and H2 - for H1 you refer to expert assessments which are not mentioned in the research, and for H2 is totally unclear

-in the conclusion you refer to existing gender barriers. You claim to have identified them, but there is no mention of those in the research

-last part in the Conclusion is contradictory and not understandable

-in the Limitations section some sentences are copied twice. The limitation referring to choice of indicators based on gender is not understandable. Why is that a limitation when you study women entrepreneurship? Also some parts of Limitations section (aim of the research) belong to conclusion and should be placed there.

Comments on the Quality of English Language

This paper needs substantial language editing - some sentences are not understandable at all and there quite a few syntax and grammar errors.

Author Response

We thank the editors and the reviewers for their interest in our work and for helpful comments that will greatly improve the manuscript. The changes suggested has been helpful and they have improved the final manuscript. The Reviewers have brought up some good points and we appreciate the opportunity to clarify the way to express some questions, and introduce details which clarify some aspects in our research.

We have tried to do our best to respond to the points raised. In accordance with your suggestions, we have undertaken a revision of the manuscript and they are in green colour.  Do not hesitate to contact us in the case that any further details or improvements are needed. We will be pleased to respond to them promptly.

Yours faithfully,

The authors

REVIEWER 2

We thank the reviewer for a comprehensive analysis of our research.

There are some  issues that need to be solved before the paper can be accepted:

-first of all, the paper needs language editing (grammar, syntax, too long and difficult to understand sentences)

Thank you. The manuscript has been sent to the editing services.

-over half of section 2.2.  talks about global issues in women entrepreneurship, while, according to the title is should concentrate on situation in Ukraine

Women's entrepreneurship in Ukraine has many problems and gender aspects that are common to most countries in the world. Therefore, we have devoted part of this section to an overview of this issue in general. Particular attention was paid to the development of women's entrepreneurship in Ukraine during the war.  Therefore, these studies can be valuable for substantiating the directions of women's entrepreneurship development in crisis conditions. In the process of eliminating comments, we added relevant information on lines 212-232.

- regarding simulation model it would be very helpful to add some more detailed explanations which you refer to later in the conclusion

More detailed explanations have been added in lines 408-425

-it is not very clear how you confirm H1 and H2 - for H1 you refer to expert assessments which are not mentioned in the research, and for H2 is totally unclear

Hypothesis H1 was formed on the basis of theoretical research, analysis of reports of global comparative systems [32,33,34,35] and analytical studies of the development of Ukrainian women's entrepreneurship during the war [50,61].

Hypothesis H2 was formed on the basis of the analysis of statistical data sets, while collecting information for correlation and regression analysis and simulation modelling.

The result of the hypothesis has been included indicating specifically if they are validated and how in the case of H1.

-in the conclusion you refer to existing gender barriers. You claim to have identified them, but there is no mention of those in the research

We agree the statement does not appropriate and this issue has been rewritten (551-553)

-last part in the Conclusion is contradictory and not understandable

The conclusion have been rewritten in the lines 516-522 and practical implications has been added in order to improve this section.

-in the Limitations section some sentences are copied twice. The limitation referring to choice of indicators based on gender is not understandable. Why is that a limitation when you study women entrepreneurship? Also some parts of Limitations section (aim of the research) belong to conclusion and should be placed there.

The Limitations has been edited and improved. Lines 559-570.

Round 2

Reviewer 2 Report

Comments and Suggestions for Authors

The paper is significantly improved. As such it can be published.

Author Response

Thank you, your suggestions have improved the manuscript in the current version.